# Predicting Object Interactions with Behavior Primitives: An Application in Stowing Tasks

**Haonan Chen, Yilong Niu,\* Kaiwen Hong,\* Shuijing Liu, Yixuan Wang,**
**Yunzhu Li, Katherine Driggs-Campbell**
University of Illinois, Urbana-Champaign
{haonan2, yilongn2, kaiwen2, sliu105, yixuan22, yunzhuli, krdc}@illinois.edu

**Abstract:** Stowing, the task of placing objects in cluttered shelves or bins, is a common task in warehouse and manufacturing operations. However, this task is still predominantly carried out by human workers as stowing is challenging to automate due to the complex multi-object interactions and long-horizon nature of the task. Previous works typically involve extensive data collection and costly human labeling of semantic priors across diverse object categories. This paper presents a method to learn a generalizable robot stowing policy from predictive model of object interactions and a single demonstration with behavior primitives. We propose a novel framework that utilizes Graph Neural Networks to predict object interactions within the parameter space of behavioral primitives. We further employ primitive-augmented trajectory optimization to search the parameters of a predefined library of heterogeneous behavioral primitives to instantiate the control action. Our framework enables robots to proficiently execute long-horizon stowing tasks with a few keyframes (3-4) from a single demonstration. Despite being solely trained in a simulation, our framework demonstrates remarkable generalization capabilities. It efficiently adapts to a broad spectrum of real-world conditions, including various shelf widths, fluctuating quantities of objects, and objects with diverse attributes such as sizes and shapes.

**Keywords:** Robotic Manipulation, Model Learning, Graph-Based Neural Dynamics, Multi-Object Interactions

## 1 Introduction

Stowing, defined as relocating an object from a table to a cluttered shelf, is one of the dominating warehouse activities. In stowing, an agent is required to pick up an object from a table. The agent must then actively create free space within the shelf before inserting the object from the table. A successful stow execution is characterized by the placement of all objects with poses in some predefined thresholds. While stowing can be performed effortlessly by humans, it remains challenging when automated with robots. The difficulty stems from the long-horizon nature, multi-object interactions, and the variety of objects and configurations involved in stowing tasks.

The challenge of long-horizon stowing task is not only due to the nature and variety of objects involved but also due to several inherent constraints in existing methods. First, the nature of these tasks requires determining the characteristics of contacts and frictions, which is a task that presents considerable difficulties. Conventional first-order models fall short in capturing the physical effects, and identifying the specific parameters of contact and friction is challenging [1]. Thus, designing a controller for such tasks becomes a tedious and laborious process. Second, the existing methodologies, including manually pre-programmed systems and recent advancements in category-level object manipulation, exhibit notable limitations. Classical pre-programmed systems struggle with adaptability, unable to efficiently handle variations introduced by different arrangements of objects on the

---

*Equal contribution

7th Conference on Robot Learning (CoRL 2023), Atlanta, USA.

shelf. Meanwhile, recent strategies for category-level object manipulation are curbed by the need for expensive data collection and human labelling, thus failing to provide a scalable solution [2]. Additionally, pure learning-based methods, such as Deep Reinforcement Learning (DRL), also present drawbacks in terms of extensive training time and poor data efficiency [3, 4], making learning from scratch on real robots impractical for long-horizon tasks.

To address these challenges, we propose a framework that uses Graph Neural Networks (GNNs) to predict object interactions within the parameter space of behavior primitives. When trained with various situations in the simulator, GNNs can learn to model the forward dynamics associated with interactions of rigid objects. Instead of explicitly determining contacts and frictions, our GNN framework is designed to learn these underlying interactions during the training process. Thus, we eliminate the need for explicit detection and intricate calculations related to contacts and forces. Our framework also applies primitive-augmented trajectory optimization to search the parameters of a predefined library of heterogeneous skills or behavior primitives. The incorporation of behavior primitives enables our policy to handle tasks of significant complexity with improved efficiency.

We make three key contributions: (1) We introduce a novel model-based imitation learning framework to learn from minimal demonstrations, which enables robots to acquire complex skills. (2) We create a comprehensive stowing benchmark for long-horizon manipulations, which is highly prevalent in various both the industrial and household applications. (3) We demonstrate the effectiveness and generalization of our framework across a wide range of real-world experiments in stowing tasks.

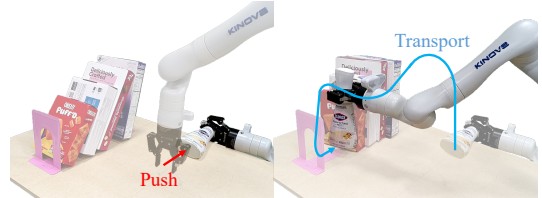

Figure 1: The robot places an object into a cluttered shelf by exploiting object interactions. It uses the grasped object to manipulate other objects within the shelf through pushing and sliding actions and finally places all the objects in the appropriate location.

## 2  Related Works

**One-shot Imitation Learning in Manipulation:** Recent advancements in imitation learning aim to solve unseen tasks with minimal demonstrations [5, 6, 7, 8, 9]. Typical approaches use a two-phase pipeline: a meta-learning phase to train a general policy from numerous tasks and a fine-tuning phase to optimize the policy for a specific task using a single demonstration [9]. Certain replay-based methods employ a strategy of estimating a 'bottleneck pose', splitting the trajectories into two segments, and then replaying the demonstration trajectory from the bottleneck [6]. Other techniques emphasize learning object representations and identifying correspondence between specific objects [5, 10] or objects within the same category [8]. However, these methods primarily handle relatively short-horizon tasks and face difficulties in modeling object dynamics.

**Model Learning in Robotic Manipulation:** Dynamic models have emerged as crucial components in robotic systems, with the capability to predict future world states based on current states and actions [11, 12, 13, 14, 15, 16]. These models can handle a variety of input representations, ranging from images and latent vectors to point sets and graph representations. Notably, graph representations have shown an exceptional ability to capture interactions among objects [17, 18, 19, 20, 21, 22, 23]. The ability to model interactions between various objects and the robot's end effector is pivotal for our research, leading us to use a graph to model interactions between objects in our system. Tekden et al. [24, 25] introduce a graph-based modeling approach that places emphasis on object-level representation, leveraging GNNs to capture interactions among multiple objects. Similarly, RD-GNN [26] uses an MLP for action encoding and treats each object as a unique node in the graph, concentrating on inter-object relations. While both approaches provide broad perspectives on object interactions, our framework diverges by representing robot movements as point clouds and objects with multiple particles. This approach offers a more granular understanding of actions and interactions, enhancing the accuracy of object movement predictions.

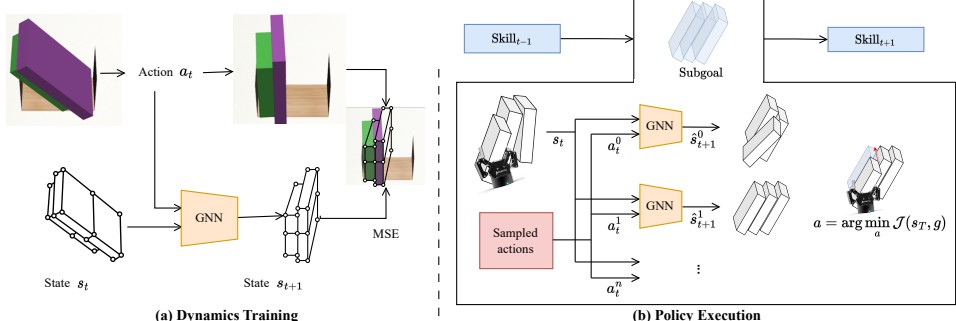

| (a) Dynamics Training | (b) Policy Execution |

Figure 2: **Overview of the proposed framework.** (a) A particle-based representation characterizes the object state. The object state's predicted outcome following the executed robot actions is computed alongside the ground truth object state using the MSE loss function to train the GNN. (b) For each skill, we apply random shooting to sample parameters within the action parameter space, utilizing the GNN to predict object movement. We then select the action that brings us closest to the desired state. Each skill is executed in sequence.

**Long-Horizon Planning in Robotic Manipulation:** Addressing long-horizon manipulation problems presents considerable complexity. Hierarchical reinforcement learning (HRL) approaches address this issue by using a high-level policy to select optimal subtasks and low-level policies to execute them [27]. However, these methods face the challenges of the sim2real gap and the difficulty of real-world data collection, hampering their real-world transferability [28]. An alternative approach by Di Palo et al. [29] augments replay-based methodologies by integrating primitive skills for routine long-horizon tasks, though their task configurations lack versatility. Integrated task and motion planning (ITMP) combines discrete and continuous planning[30], blending high-level symbolic with low-level geometric reasoning for extended robotic action reasoning. Recent efforts by Lin et al. [31] have explored sequential manipulation of deformable objects through a differential simulator and trajectory optimization. However, this work is only validated in simulation, and real-world deployment is non-trivial due to the difficulty of obtaining gradients of state changes. To address this, we propose to apply trajectory optimization to GNN-based forward dynamics prediction modules, incorporating heterogeneous behavior primitives.

## 3   Approach

In our proposed system, a GNN first predicts system dynamics. Then, a primitive-augmented trajectory optimization method achieves subgoals from a single demonstration, which is shown in Figure 2. Initially, the object state is represented as particles. We then train the GNN with the MSE loss between the predicted outcome following robot actions and the ground truth state. We use the random shooting to explore the action parameter space and use the GNN's predicted results to select the optimal action that aligns most closely with our desired state. The skills are executed in a sequential manner, guiding our system to accomplish its tasks efficiently.

### 3.1   Learning Forward Dynamics via GNN

**Graph Construction:**   We define a dynamics function that describes a rigid-body system state $\mathcal{S}$ with $M$ objects and $N$ particles. We model each rigid object as a collection of uniformly distributed particles, offering a representation that is flexible and adaptable to variations in object shape, size, and geometry. The dynamics function is expressed as $\Phi : \mathcal{S} \times \mathcal{A} \to \mathcal{T}$. $\mathcal{A}$ represents the skill type and its associated parameters, and $\mathcal{T}$ denotes $M$ rigid transformations containing the translation and rotation for each dynamic object. The future state of an object can be determined by applying a sequence of these rigid transformations. We represent each rigid objects as uniformly distributed particles as such representations are versatile to object shape, object size, and object geometry.

Each object in the system is represented by its respective particles. We define the graph state $s_t = (\mathcal{O}_t, \mathcal{E}_t)$, where the graph's vertices $\mathcal{O}_t$ represent an object's particles, and edges $\mathcal{E}_t$ denotes the relations between particles. Each vertex $o_{i,t} = \langle x_{i,t}, c_{i,t} \rangle$ consists of the particle's position $x_{i,t}$

and object's attributes $c_{i,t}$ including its dynamism (dynamic or static), the object it belongs to, its offset from the rigid body's center, and gravity. Edges $\mathcal{E}_t$ are formed when the distance between two vertices is less than a specified threshold. In our work, the relations are characterized by the physical interactions and geometric proximity between the particles signifying objects. Specifically, we introduce the following relations to adeptly encapsulate the complex dynamics in multi-object interactions: (1) **Intra-object relations**: Between different particles within the same object or across different objects. (2) **Gripper-to-object relations**: Between particles from the objects and the robot's gripper. The edge relations are represented as $e_k = \langle i_k, j_k, c_k \rangle$, in which $i_k, j_k$ denote the indices of the receiver and sender particles, respectively. The edge index is denoted by $k$, and nature of the relationship such as intra-object relation, or gripper-to-object relation) is denoted by $c_k$. Since our focus is predicting the motions of dynamic objects, we restrict node building to vertices associated with these dynamic objects. However, to effectively model the interactions between dynamic and static objects (i.e, shelf and table), we choose to incorporate the particles of static objects during the construction of edges.

**Message Passing:** The features of all vertices and edges are processed through encoder multi-layer perceptrons (MLPs), resulting in latent vertex representations denoted as $h_i^O$ and latent edge representations represented as $h_{i,j}^E$. At each time step, we apply the following message function to the vertex feature and edge feature for several iterations to handle signal propagation.

$$h_{i,j}^E \leftarrow \rho^E(h_{i,j}^E, h_i^O, h_j^O), \quad h_i^O \leftarrow \rho^O(h_i^O, \sum_j h_{i,j}^E). \tag{1}$$

where the message passing functions $\rho^E$ and $\rho^O$ are MLPs. Subsequently, we apply an MLP decoder to the vertex feature processor's final layer output. The rigid transformation for each individual object is determined by calculating the mean of the decoder outputs corresponding to that object.

**Representing Action as Particles:** The control action, represented by the gripper's planned position and motion, is defined by particles $o_{i,t} = \langle x_{i,t}, v_{i,t}, c_{i,t} \rangle$, where $x_{i,t}$ denotes the current position of gripper, $v_{i,t}$ denotes the planned motion of the gripper. The particles associated with the gripper are subsequently encoded by the encoder. Additionally, we predict the future positions of the gripper. The discrepancy between these predicted positions and the actual achieved positions serves as an auxiliary loss, which helps GNNs better understand the inherent physical laws and constraints.

## 3.2 Control with the Learned Dynamics

In this section, we discuss the design of behavior primitives, and trajectory optimization algorithms used to generate parameters for different skills.

**Behavior Primitives:** We introduce the behavior primitives as a higher level of temporal abstraction to facilitate efficient planning. The behavior primitives simplify the task space by generating key poses for the system, which subsequently executes actions using Operational Space Control (OSC) [32] as a lower-level controller. The GNN is used only to predict the system's state at the key poses of behavior primitives, which significantly reduces the number of forward passes and cumulative prediction error over time. Behavior primitives function serve as building blocks and can be easily extended for various manipulation tasks. We further specify a maximum execution time $T_{skl}$ for each behavior primitive. Our system integrates a collection of three primitives, encompassing both prehensile and non-prehensile motions. The primitives and their associated parameters are described as follows: (1) **Sweeping:** The robot sweeps objects on the shelf using its end-effector, aiming to stand them upright. Sweeping is parameterized by the starting offset $y$ in the shelf direction, sweeping height $h$, sliding distance $d$, and the angle of gripper rotation $\theta$ during the sweep. (2) **Pushing:** Pushing involves the robot nudging the object to establish potential grasp poses. The starting push position $(x, y)$ and the distance of the push $d$ are the parameters for pushing. (3) **Transporting:** The robot picks up the object from the table, places it in the shelf, and, if necessary, adjusts its position through sliding. This skill is defined by parameters such as the starting offset in the shelf direction $y$, the height of insertion $h$, the sliding distance $d$, and the gripper rotation angle $\theta$ during the insertion process.

**Goal-conditioned Trajectory Optimization:** We optimize the parameters of a given skill by minimizing the mean square error (MSE) loss between the predicted particle positions and their corresponding positions as demonstrated. Keyframes collected during demonstrations can be denoted by $g$. We search for the skill parameter $a_p$ that minimizes the cost function $\mathcal{J}$ representing the MSE between the predicted and target positions of the object particles. Mathematically, this optimization problem is represented as $a_p = \arg\min_{a_p} \mathcal{J}(s_T, g)$. The resulting low-level control actions are generated from the the skill which is parameterized by the skill parameters $a_p$. Our dynamics network $\Phi$ makes forward predictions of the future state of the system after the execution of each skill. We employ trajectory optimization to find the skill parameters that yield the lowest cost.

## 4  Experiment Setup

Our experimental setup consists of both simulated and real-world environments. The simulated environment is built using Robosuite [33] and the MuJoCo physics engine [34], operating with a 7-DOF Kinova Gen3 Robot. The real-world counterpart consists the same Kinova Gen3 Robot and a Robotiq 2F-85 gripper.

**Data collection:** We collect a dataset of 300 episodes of action execution for each behavior primitive, where the actions are executed based on randomly sampled skill parameters. For each episode, the robot randomly sampled parameters within its parameter space and executed the skill in the simulator. We gather the key poses for each skill, subsequently training a GNN to take the state at these key poses and the robot action and predict the future state at the subsequent key poses.

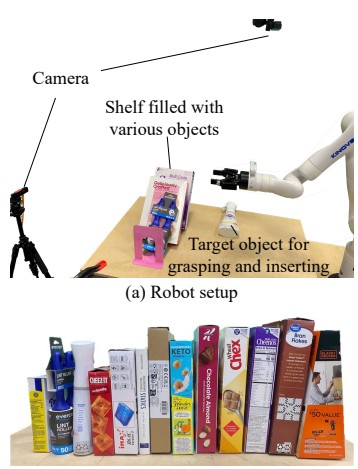

(a) Robot setup

(b) Objects used in this work

Figure 3: The experimental setup.

**Simulated environment:** In our simulation, the shelf width is initialized to randomly vary within the range from 0.18 to 0.35 meters. We also randomly select the number of objects placed on the shelf, varying between two and four. The properties of each object, including size and density, were randomly generated. All the objects are created with a box shape. We use a SpaceMouse to teleoperate the robot to complete the task, and the ending poses of each skill were collected. These poses are then used as subgoals for each skill during execution.

**Real-world environment:** Figure 3 illustrates our real-world environment setup. We use OAK-D Pro cameras, with a top-down camera to estimate the pose of the object for grasping and inserting, and a side-view camera to estimate the object's pose in the shelf. Objects placed on the table are always oriented perpendicular to the table edge and positioned adjacent to it. Shelf sizes of 0.18m and 0.35m are tested, and objects in the shelf are placed at randomized positions and orientations. A point cloud representation of each object is created, including their sizes, positions, and orientations as the state representation.

**Evaluation metrics:** We evaluated our dynamics model and manipulation outcomes in simulation based on the final prediction error, applying metrics such as Mean Squared Error (MSE), Earth Mover's Distance (EMD) [35], and Chamfer distance (CD) [36]. In real-world scenarios, success rates were computed for each setup. A success was defined as all boxes ending up within the shelf with their orientations falling within a predefined threshold $\theta$.

## 5  Experimental Results

In this section, we evaluate our forward dynamics prediction model in both simulated and real-world settings. We use a diverse range of objects and shelf dimensions to provide a broad and challenging test. The results highlight our framework's potential for zero-shot sim2real transfer, demonstrating its applicability in handling real-world conditions without the necessity for prior real-world data.

Table 1: **Dynamics Model Prediction Quantitative Results and Ablations.** Our model consistently outperforms RoboCraft in complex primitives such as 'Sweep' and 'Transport', evident in the lower MSE, EMD, and CD values. The Object-Level representation underperforms due to its lack of dense object information. In the simpler 'Push' primitive, our model retains comparable MSE, EMD, and CD values within one standard deviation, indicating robust performance even in simpler scenarios.

| Primitive | Method | MSE (mm) ↓ | EMD (mm) ↓ | CD (mm) ↓ |
|---|---|---|---|---|
| Sweep | Object-Level Repr | 3.676 (± 1.597) | 75.015 (± 14.452) | 92.914 (± 17.206) |
| | RoboCraft [14] | 0.351 (± 0.222) | 24.510 (± 6.434) | 33.277 (± 5.064) |
| | Ours | **0.287 (± 0.185)** | **21.792 (± 5.533)** | **30.017 (± 3.259)** |
| Push | Object-Level Repr | 3.765 (± 0.76) | 75.975 (± 10.927) | 95.925 (± 14.802) |
| | RoboCraft | **0.216 (± 0.148)** | **12.509 (± 3.494)** | 15.43 (± **2.902**) |
| | Ours | 0.292 (± 0.179) | 14.569 (± 3.752) | **15.046** (± 3.085) |
| Transport | Object-Level Repr | 5.861 (± 3.106) | 91.913 (± 20.552) | 113.263 (± 23.168) |
| | RoboCraft | 1.091 (± 0.512) | 42.162 (± 10.317) | 55.615 (± 9.997) |
| | Ours | **0.666 (± 0.41)** | **31.232 (± 9.108)** | **38.068 (± 6.605)** |

Table 2: **Quantitative Results of Control in Simulation.** Our method consistently outperforms SAC, PPO, parameterized PPO, and heuristic-driven control, exhibiting markedly lower execution errors, illustrating the critical role of incorporating a model for long-horizon stowing tasks.

| Method | MSE (mm) ↓ | EMD (mm) ↓ | CD (mm) ↓ |
|---|---|---|---|
| SAC | 265.411 | 465.639 | 368.571 |
| PPO | 87.479 | 266.554 | 173.522 |
| Parameterized PPO | 22.925 | 120.736 | 62.182 |
| Heuristic | 34.861 | 140.196 | 194.123 |
| Ours | **0.905** | **29.697** | **39.914** |

## 5.1 Dynamics Model Learning and Ablations

We trained our dynamics model using MSE loss. As part of our ablation study, we incorporated a version of the GNN presented in [14], which we will refer to as "RoboCraft". It's important to note that this model does not encompass dynamic-static object interactions, nor does it utilize the auxiliary loss derived from gripper movements. Additionally, we introduced a baseline, "Object-Level Repr", using object-level representation by using a single GNN node to symbolize objects.

The quantitative results from the model learning are shown in Table 1. The relatively small prediction errors suggest that our model is able to accurately predict the interactions involved in the task; these include object/object interactions, environment/object interactions, and robot gripper/object interactions. The results further demonstrate that the model can effectively learn the rigid motions of the objects, resulting in only minor errors.

Compared to RoboCraft, the introduction of gripper movement information and dynamic-static object edge information improves the prediction accuracy of our GNN model, particularly in complex behavior primitives such as sweeping and transporting. Even in simpler behavior primitives like pushing, our GNN maintains a performance comparable to the RoboCraft, with the prediction error remaining within one standard deviation, highlighting its consistent effectiveness. While the less specialized RoboCraft performs adequately in straightforward pushing skill, it encounters difficulties in more dynamic situations. In contrast, our model's advanced complexity and adaptability prove to be particularly advantageous in scenarios characterized by intricate dynamics, such as collisions and bounces between objects or with the environment.

## 5.2 Manipulation Results

**Results in Simulation:** We collect six demonstrations in the simulation. The keyframes from these demonstrations serve as the goal state for trajectory optimization, implemented with Random Shooting (RS). In our analysis, we evaluated Random Shooting, the Cross-Entropy Method, and Gradient Descent - all of which exhibited similar performance. We present the results obtained from RS and denote them as "Ours" in the subsequent discussion. We use Proximal Policy Optimization (PPO) [37] and Soft Actor-Critic (SAC) [38], both state-of-the-art, model-free RL algorithms, as baselines. The choice of these algorithms aims to highlight the necessity of a model for long-

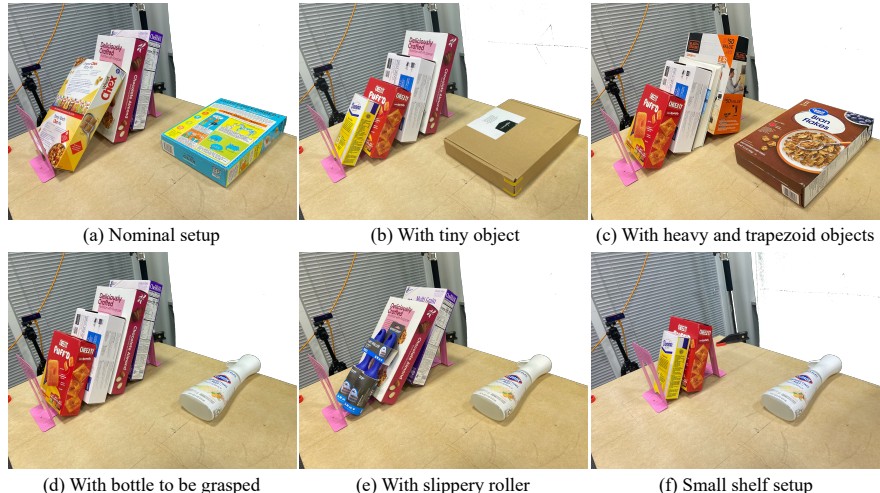


(a) Nominal setup      (b) With tiny object      (c) With heavy and trapezoid objects



(d) With bottle to be grasped      (e) With slippery roller      (f) Small shelf setup


Figure 4: **Different Setups Used in Real-World Manipulation Experiments.** Six setups represent a wide range of conditions including different combinations of objects, shelf sizes, object dimensions, and shapes.

horizon stowing tasks. The simulation results are presented in Table 2. Each method's performance is assessed using MSE, EMD, and CD as metrics. PPO and SAC utilize the negative MSE between the current and goal object states as the reward function. In comparison to these methods, RS consistently yields lower execution errors across all metrics, which indicates its superior capability in minimizing the gap between actual and desired states. Model-free RL methods such as PPO and SAC struggle to perform effectively due to the large exploration space and the long-horizon nature of the task. Their effectiveness is further hampered by their lack of knowledge regarding the model of the environment. We also benchmark against a version of PPO with a parameterized action space based on behavior primitives and a heuristic-driven approach devoid of learned dynamics. Our method outperforms both, demonstrating a considerable performance advantage.

**Results in the Real World:** Our framework is tested in six different real-world setups, with each setup executed for ten test trails. We manually randomize the initial orientations and positions of the objects within the shelf in each trail. The object poses in these scenarios are identified using Scale-Invariant Feature Transform (SIFT) on the captured images. The various setups represent a broad spectrum of conditions including different object combinations, shelf sizes, object dimensions, and shapes. This wide range of conditions is depicted in Figure 4.

In our experiments, we implement two distinct skill combinations for each setup: a 3-skill set comprising sweeping, pushing, and transporting, and a 2-skill set, which only included pushing and transporting. The term "heuristic" refers to a process where humans fine-tune a relatively small parameter space and assign the tuned parameters to the skills. The heuristic-based approach is likewise conducted utilizing all three skills: sweeping, pushing, and transporting.

Table 3: **Real-World Success Rates.** Performance evaluation in six different setups using: our proposed method with two distinct skill sets (2 skills and 3 skills), RoboCraft as learned dynamics, and a heuristic-based approach without dynamics prediction.

| Setups | Success ↑ | | | |
|---|---|---|---|---|
| | **Heuristic** | **RoboCraft** | **3 skills** | **2 skills** |
| (a) | 1/10 | 4/10 | 10/10 | 10/10 |
| (b) | 3/10 | 7/10 | 9/10 | 9/10 |
| (c) | 3/10 | 4/10 | 9/10 | 9/10 |
| (d) | 1/10 | 6/10 | 10/10 | 10/10 |
| (e) | 2/10 | 7/10 | 10/10 | 10/10 |
| (f) | 1/10 | 5/10 | 9/10 | 9/10 |
| Average | 18% | 48% | 95% | 95% |

Figure 4 presents the success rates of the different control strategies. Our 3-skill method significantly outperforms the heuristic-based approach with a success rate of 95%, indicating the effective handling of varied setups by our dynamics prediction module. Interestingly, the 2-skill set also achieves the same 95% success rate, indicating that the robot's ability to understand the interactions between the gripper-held object and the objects within the shelf enables it to determine the optimal position for insertion and placement. These high success rates demonstrate the effectiveness of our method.

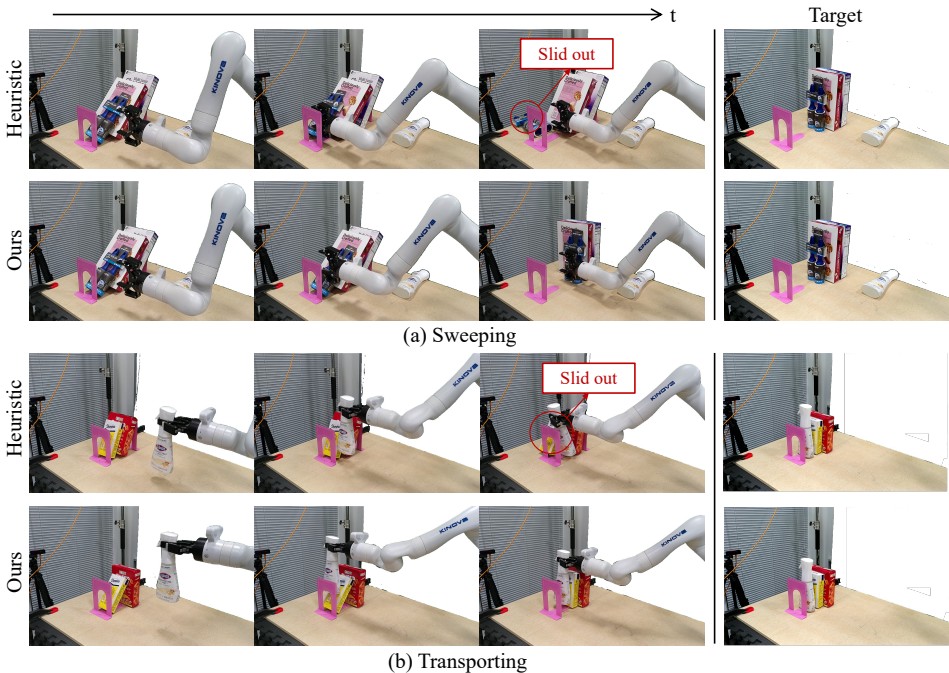

(a) Sweeping

(b) Transporting

Figure 5: **Comparison of the heuristic-based method and our approach during the execution of the sweeping skill and transporting skill.** Our method anticipates future states and arranges objects into upright positions within the shelf, unlike the heuristic-based method which pushes objects out of the shelf.

In contrast, the heuristic-based strategy yielded an average success rate of only 18%. Despite being trained solely with box-shaped objects, our method generalizes effectively to out-of-distribution objects, showing its versatility in a variety of real-world conditions.

Figure 5 provides a qualitative comparison of the sweeping skill execution between the heuristic-based method and our approach. Our method, equipped with the ability to anticipate future states based on specific robot actions, is capable of sweeping and transporting objects into upright positions within the shelf. In contrast, the heuristic-based method tends to push objects out of the shelf.

## 6 Conclusion

In this work, we focus on stowing tasks wherein a robot must manipulate a large, ungraspable flat object and subsequently stow it within a cluttered shelf. The robot's task involves creating sufficient space within the cluttered shelf to accommodate the object appropriately. We introduce a system that utilizes behavior primitives in combination with forward dynamics prediction to achieve the task. We discuss the design choices for our system and demonstrate its effectiveness in real robot scenarios. Moreover, we show the system's ability to generalize to various stowing conditions.

Our work opens several potential avenues for future research. One promising direction involves developing the ability to composite skills and further reduce the sub-goals presented in the demonstrations. Another is that the design and definition of the behavior primitives library need additional exploration and research, which can enhance the adaptability and versatility of robotics systems in performing complex manipulation tasks.

**Limitations**: Our system currently has a few limitations. Firstly, it relies on manual human labeling of ordered keyframes from demonstrations, which could potentially restrict scalability and deployment in larger and more complex scenarios. Secondly, we use box-shaped point clouds to represent objects during training and inference. This simplistic representation may not accurately reflect the geometrical properties of objects, especially in scenarios involving more complex interactions and contacts. Addressing these limitations, particularly improving object representation, presents a promising direction for future research.

**Acknowledgments**

We thank Haochen Shi's tireless assistance with the GNN implementation, as well as Neeloy Chakraborty, Peter Du, Pulkit Katdare, Ye-Ji Mun, and Zhe Huang for their insightful feedback and suggestions.

This work was supported by ZJU-UIUC Joint Research Center Project No. DREMES 202003, funded by Zhejiang University.

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

# Appendices

## A  Additional Results

### A.1  Generalization to Deformable Objects in Real-World Experiments

In an effort to explore the adaptability of our method, we further extended our experiments to consider deformable objects. Intriguingly, even though our methodology was trained predominantly using box-shaped rigid objects, it showcased commendable generalization capabilities. Nevertheless, as expected, the intricacies of deformable objects' dynamics, being distinct from rigid ones, posed challenges, resulting in somewhat diminished performance compared to other test cases.

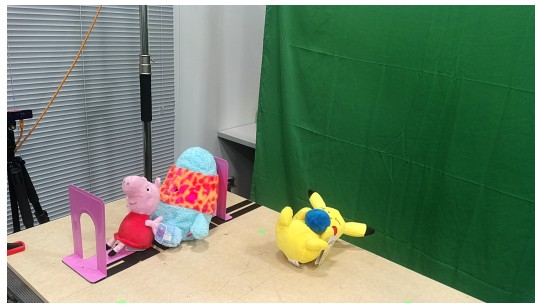

Figure 6: Experimental setup with deformable objects.

|  | Success Rate ↑ | | | |
|---|---|---|---|---|
| **Setups** | **Heuristic** | **RoboCraft** | **3 skills** | **2 skills** |
| Deformable Objects | 0/10 | 2/10 | 6/10 | 4/10 |

Table 4: Success rates when dealing with deformable objects.

### A.2  Impact of Sample Size on Prediction Performance

We examined the influence of training sample size on dynamics prediction performance. By varying the number of training samples and maintaining a consistent validation and test set, we observed the relationship between sample size and prediction accuracy in Figure 7. Specifically:

- For the **push** behavior primitive, performance improvements plateaued around 50 samples.
- For the **sweep** behavior primitive, a similar plateau was observed at 50 samples.
- The **transport** behavior primitive exhibited steady performance gains until approximately 140 samples.

## B  Implementation Details

### B.1  Determining the Edge Formation Threshold in GNN

The optimal threshold for forming an edge in our graph representation was empirically derived to strike a balance between computational complexity and representational expressiveness. Just as with other hyperparameters essential for training our deep neural networks, we determined thresholds through extensive experimentation, primarily guided by validation loss. Depending on the specific behavior being modeled, we adopted the following thresholds to ensure the edges encapsulate meaningful interactions:

**Transport:**

- Intra-object relations: 0.15m
- Gripper-to-object relations: 0.175m

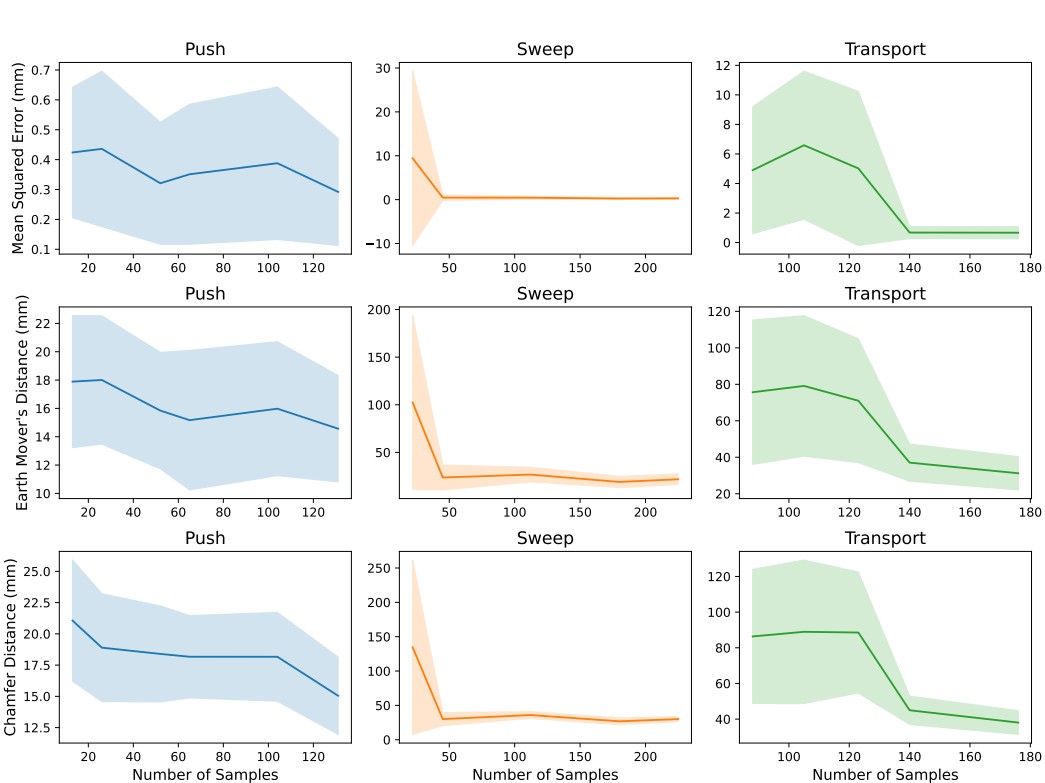

Figure 7: Variation in prediction performance as a function of training sample size.

**Push:**

- Intra-object relations: 0.1m
- Gripper-to-object relations: 0.1m

**Sweep:**

- Intra-object relations: 0.175m
- Gripper-to-object relations: 0.175m

When the contact distance is set excessively large, it leads to the inclusion of a significant number of edges that may not be relevant to the scene, which makes the model computationally expensive and introduces noise which can hinder both the training process and the quality of inference. A threshold that is too small may miss vital interaction relations. We tuned these thresholds to effectively capture the nuances of object interactions to optimize the prediction accuracy.

### B.2 Real Robot Experiment

**Perception:** Our perception pipeline uses two OAK-D Pro cameras to estimate the 6D (x, y, z, roll, pitch, yaw) pose of objects. The side camera, oriented towards the shelf, determines the poses of the objects on the shelf. In contrast, the top camera determines the pose of the object situated on the table. We maintain a database encompassing comprehensive details of all objects in the experiment, including their size and texture. We approximate all objects as cuboids, defined by three parameters: length, width, and height. Side and top view images of these objects are captured to form a collection of ground truth images. These images aid the identification of objects via the SIFT (Scale Invariant Feature Transform) feature matching algorithm. This algorithm is executed for each object in the

scene to match keypoints, which are subsequently used to calculate the homography matrix. We then use this matrix to calculate the 2D coordinates and 1D orientation for each object in the image space. Given the setup of our experiment, it suffices to estimate 3 degrees of freedom (DOF) for all objects: (y, z, roll) for objects on the shelf and (x, y, yaw) for the object on the table. The other 3 DOF is deterministic based on the known shelf location and object size. Incorporating the 3D pose of each object in the image space with our prior knowledge of the experiment setup, we can obtain the 6D pose for all the objects in the scene for robot manipulation.

 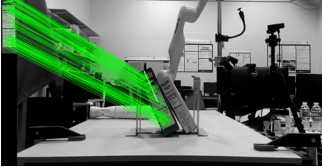 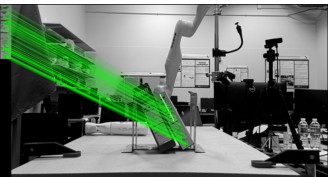

(a) SIFT featuring matching with top-down view camera

(b) SIFT feature matching with side view camera

(c) SIFT feature matching with side view camera

Figure 8: SIFT results with setup-(f). The ground truth images collected before the experiment are shown in the upper-left corner of each subfigure. The green lines show the keypoints correspondence.

## B.3 More information on the objects

Table 5: Object Dimensions

| Object | Length (m) | Width (m) | Height (m) | Image |
|---|---|---|---|---|
| Sugar box | 0.089 | 0.040 | 0.177 | |
| Lint roller | 0.110 | 0.050 | 0.230 | |
| Clorox bottle | 0.065 | 0.255 | 0.055 | |
| Cheez-it box | 0.152 | 0.047 | 0.190 | |
| Charger box | 0.168 | 0.051 | 0.240 | |
|  | | | | |

| Object | Length (m) | Width (m) | Height (m) | Image |
|---|---|---|---|---|
| Book | 0.203 | 0.028 | 0.245 |  |
| Amazon box | 0.210 | 0.288 | 0.045 |  |
| Keto box | 0.193 | 0.259 | 0.046 |  |
| Kellogg's box | 0.190 | 0.045 | 0.284 |  |
| Chex box | 0.195 | 0.050 | 0.285 |  |
| Cheerios box | 0.195 | 0.052 | 0.285 |  |
| Bran flakes box | 0.198 | 0.297 | 0.057 |  |
| Power drill box | 0.250 | 0.073 | 0.279 |  |
| Yellow plush toy | 0.065 | 0.255 | 0.055 |  |
| Blue plush toy | 0.19 | 0.284 | 0.045 |  |

| Object | Length (m) | Width (m) | Height (m) | Image |
|--------|------------|-----------|------------|-------|
| Red plush toy | 0.168 | 0.24 | 0.051 | |

## B.4  Subgoal Selection for Behavior Primitives

Figure 9 illustrates the typical subgoals selected during our real-world experiments. In our approach, subgoals serve as intermediate waypoints or milestones within the task. They provide a sequence of specific, short-term objectives for the robot to achieve, ultimately guiding it towards the final desired outcome of the task. This structure helps in breaking down a complex, long-horizon task into more manageable segments, making it easier for the robot to execute and adapt.

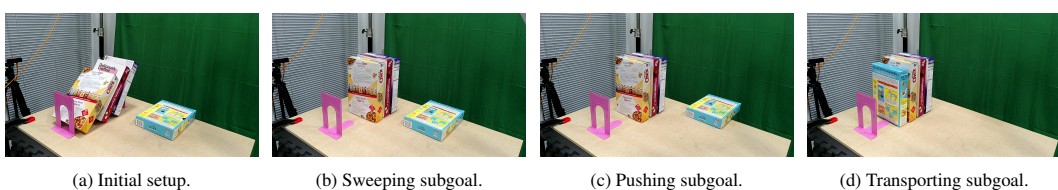

(a) Initial setup.          (b) Sweeping subgoal.          (c) Pushing subgoal.          (d) Transporting subgoal.

Figure 9: Typical subgoals selected during experiments. From left to right: (a) Initial setup, (b) Sweeping, (c) Pushing, (d) Transporting.

