# OpenReview forum: "Predicting Object Interactions with Behavior Primitives: An Application in Stowing Tasks"
_robot-learning.org/CoRL/2023/Conference — CoRL 2023 Oral_

### Official Review · Reviewer_pqt2 · 2023-07-06

**Confidence:** 4
**Originality:** Good
**Technical Quality:** Good
**Clarity Of Presentation:** Fair
**Impact:** 3

**Recommendation:**

Weak Accept: I recommend accepting the paper, but will not argue for my recommendation if the majority of other reviewers have a different opinion.

**Review:**

Strengths of the paper:
1: The stowing task is novel and important for the robot manipulation community.

2: This approach is trained purely in simulation and can generalize to real-world scenarios.

3: The real-robot demos look cool and interesting.

Weakness of the paper:
1: the author proposes a graph neural network framework to model the dynamics of multi-object interactions. Both dynamics learning and planning are pretty similar to [1]. The authors should discuss the difference with the previous work.

2: The author highlights the contribution of long-horizon planning but does not cite enough related works like task and motion planning [2].

3: The paper is hard to follow. For example, the authors define subgoals at the beginning of the approach section without defining them. Also, the authors do not discuss the details of the loss function. If it’s limited by space, the authors should at least put the details of the loss function in the appendix.

4: Planning is a very important part of the paper but the authors do not discuss that part clearly. The authors only mention “We employ trajectory optimization to find the skill parameters that yield the lowest cost.” How do you optimize both the discrete part and continuous part of your skill primitives?

5: The authors claim “In our analysis, we evaluated Random Shooting, the Cross-Entropy Method, and Gradient Descent - all of which exhibited similar performance.” This seems weird to me, it’s clear that the cross-entropy method and gradient descent will perform much better than naive random shooting. My guess is that the assumptions of manually choosing good keyframes make the planning problem easy so even the naive random shooting works well.

Minor points:
1: Figure 2 is hard to understand.

2: training time instead of training times.

[1] Yixuan Huang, Adam Conkey, and Tucker Hermans. Planning for Multi-Object Manipulation with Graph Neu- ral Network Relational Classifiers. In IEEE International Conference on Robotics and Automation (ICRA), 2023.

[2] C. R. Garrett, R. Chitnis, R. Holladay, B. Kim, T. Silver, L. P. Kaelbling, and T. Lozano-Perez, “Integrated task and motion planning,” Annual review of control, robotics, and autonomous systems, vol. 4, pp. 265–293, 2021.

**Quality Of The Limitations Section:**

Limitations are addressed clearly

**Questions For Rebuttal:**

How does the author choose the keyframes or subgoals?
This seems like a very important assumption during both training and planning.

More details about the planning part. How do you optimize both the discrete part and continuous part of your skill primitives?

**Robotics Focus:**

Sufficient demonstration on hardware

**Summary Of Paper:**

This paper proposes a graph neural network framework to address the task of stowing. Also, given the demonstration during training, this paper trains a graph neural networks-based dynamics model to model the interaction of multiple objects. With the learned dynamics, the paper uses a random shooting to choose the best action to achieve several keyframes sequentially to achieve the goal. Based on the training purely in simulation, this framework can generalize to the real-world without any finetuning.

**Summary Of Recommendation:**

This paper investigates a novel and interesting stowing task with some cool real-robot demos. However, this paper should discuss more related works like [1][2] to highlight the core contributions of this paper. Furthermore, the presentation of the paper is unclear and the paper should include some important details about the planning algorithm. Thus I recommend “weak reject”.

---

### Official Review · Reviewer_kz8m · 2023-07-15

**Confidence:** 5
**Originality:** Good
**Technical Quality:** Good
**Clarity Of Presentation:** Excellent
**Impact:** 3

**Recommendation:**

Weak Accept: I recommend accepting the paper, but will not argue for my recommendation if the majority of other reviewers have a different opinion.

**Review:**

- The paper is very well-written, easy to follow.
- The idea of using GNN for predictive modeling of multiple object interactions is not new, as the authors indicated, but the authors missed some very related work [Tekden, 2020; Tekden 2021]. The proposed method uses particle-based object representation, whereas Tekden uses object-level representation in GNN. As all objects are assumed to be box-shaped rigid objects, does more-complex particle-based representation have any advantage?
- As a general remark, although I believe that the framework is quite general, the paper writing and method description is too much geared towards the stowing task, which is, while very common, a specific task among many others. The authors should provide a more general approach.
- The authors mention some challenges related to detecting contacts and friction, but they do not come back to these challenges in the rest of the paper.
- The argument in lines 38-39 (related to failure due to human labeling) needs reference and justification. Most of this work is not self-supervised.
- The argument in lines 83-84 also needs justification/reference.
- Please explain what Random Shooting corresponds to.
- In 103-104, the authors argue that their approach is flexible and adaptable to variations in object shape, size, and geometry. This is not rigorously justified in the experiments section. This can be shown by verifying the learned system with objects not observed during training.
- The sub-goal explanation for Fig. 2 (l95) is not clear.
- l113: The attributes include gravity; probably, it is supposed to be either density or mass.
- l114: How did you set the threshold to form an edge?
- l116: Please provide the complete set of relations.
- l112: What does 'object it belongs to' mean? Do objects have ids?
- The goal-conditioned trajectory optimization part is not clear. Please precisely explain the algorithm. How exactly the optimization occurs?

- How is the data, collected in data collection, being used precisely? To train GNNs? Are 300 episodes sufficient to cover the wide range of object-action interactions?
- How is the data obtained from the demonstrations used exactly? The authors mentioned they used teleoperation. How many teleoperations are done?
- In the explanation of real-world experiments, point cloud representation is indicated to be created. How is this information used?

- In the first experiment, the proposed method is shown to outperform the GNN-based baseline. However, the margin is small, and they are very similar methods with slight differences. Therefore the contribution of this result is not substantial.
- In the second experiment, the proposed method is compared against a completely different approach, RL-based methods, where no details are provided. The authors need to compare theirs with an MPC-based method to show effectiveness.
- How is the randomization realized in the third experiment (in the real world)? The assumption of complete object knowledge is a very strong one, and together with the object recognition step, the details should be moved from the appendix to the main manuscript. The baseline 'heuristic' approach is not clearly explained, and from the videos, it looks not a strong baseline.


Tekden et al. "Belief regulated dual propagation nets for learning action effects on groups of articulated objects." ICRA 2020.
Tekden et al. "Object and relation centric representations for push effect prediction." arXiv preprint arXiv:2102.02100 (2021).

**Quality Of The Limitations Section:**

Additional details required

**Questions For Rebuttal:**

- Please provide the novelty of the prediction part of this work considering other GNN-based predictors.
- Please provide a comparison of the complete system with a more similar framework, for example, Tekden et al.

**Robotics Focus:**

Sufficient demonstration on hardware

**Summary Of Paper:**

This paper implements a pipeline that predicts object pose changes using Graph Neural Networks (GNN) in an environment with a variable number of objects. The system is designed to control the robot gripper, minimizing the expected and actual object pose in order to successfully execute a number of skills, including sweeping, pushing, grasping-transferring-releasing an object. For this, the system needs to learn GNN-based effect prediction on objects given action parameters, i.e. typical motions of objects in response to the actions. These are learned from random executions and human demonstrations. In the use case, the actions are chained in order to stow an object to a rack that includes other objects. The learning occurred in the simulator and was verified in the simulator and the real world. The authors provided results and comparisons with other methods.

**Summary Of Recommendation:**

The paper proposed a sound method that combines an existing GNN-based method with a simple action control optimizer. While I appreciate the writing, the working system, and the verification in the simulator and real world, the paper does not make a strong contribution to the field theoretically or as a very strong system in practice.

Update: The paper now shows that their particle-based GNN outperforms the existing object-level GNN architectures. The paper now also includes deformable objects in experiments. Therefore, I increase my recommendation score as the paper now justifies its contribution better.

---

> ### Author Response · Authors · 2023-08-15
> **Response to Reviewer kz8m (part 2/5)**
>
> > Q: l114: How did you set the threshold to form an edge?
>
> The threshold for forming an edge in our graph representation was determined empirically to balance complexity and representational power. Through experimentation, we set thresholds based on validation loss --- similar to all other hyperparameters used in training the deep neural networks.
> Specifically, we chose the following thresholds based on the specific behavior being modeled to ensure that the edges represent significant interactions:
>
> Transport:
>
> - Intra-object particle edges: 0.15m
> - Edges between object particles and gripper particles: 0.175m
>
> Push:
>
> - Intra-object particle edges: 0.1m
> - Edges between object particles and gripper particles: 0.1m
>
> Sweep:
>
> - Intra-object particle edges: 0.175m
> - Edges between object particles and gripper particles: 0.175m
>
>
> > Q: l116: Please provide the complete set of relations.
>
> In our work, the relations are defined based on the physical interactions and geometric proximity between the particles representing objects. Specifically, we consider the following relations:
> Intra-object relations: Between different particles within the same object or across different objects.
> Gripper-to-object relations: Between particles from the objects and the robot's gripper.
> These relations enable our model to effectively capture the intricate dynamics involved in multi-object interactions. We appreciate your interest and will ensure to include a more detailed description of these relations in our revised manuscript.
>
> > Q: l112: What does 'object it belongs to' mean? Do objects have ids?
>
> In our model, objects are represented by multiple particles, and each particle carries a one-hot vector attribute to identify the object to which it belongs. The term "ids" refers to these identifiers, effectively serving as indexes for different objects. This granular representation allows us to capture complex interactions by associating particles with specific objects through these identifiers.
>
> > Q: The goal-conditioned trajectory optimization part is not clear. Please precisely explain the algorithm. How exactly the optimization occurs?
>
> The goal-conditioned trajectory optimization in our work focuses on optimizing the execution of primitive skills to align with a task objective. The trajectory optimization involves the following steps:
>
> **Primitive Selection**: A primitive is chosen based on the current state and goal to align with the demonstration.
>
> **GNN Prediction**: The GNN is used to predict the future state after executing the selected primitive with an initial set of parameters.
>
> **Optimization Objective**: We aim to minimize the mean square error between the GNN-predicted future state and the desired goal state.
>
> **Parameter Optimization**: We refine the primitive parameters to minimize the discrepancy between the predicted and desired states.
>
> **Execution**: The optimized parameters are used to execute the primitive, aligning the system with the goal.
>
> This trajectory optimization procedure ensures that the system's actions are guided by the desired goal state, leading to more precise and goal-aligned primitive execution.

---

> > ### Author Response · Authors · 2023-08-15
> > **Response to Reviewer kz8m (part 3/5)**
> >
> > > Q: How is the data, collected in data collection, being used precisely? To train GNNs? Are 300 episodes sufficient to cover the wide range of object-action interactions?
> >
> > **Data Usage and GNN Training:**
> >
> > In our method, we randomly sample parameters within each primitive's parameter space and capture the state, action, and next state. This collected data is used to train our forward prediction GNNs. Each primitive's dataset consists of around 300 episodes that encapsulate diverse object-action interactions. Experimental validation has underscored our model's adeptness in predicting object interactions.
> > Furthermore, as we vary the training dataset size while keeping the validation and testing sets consistent, we observed that performance reached a plateau before hitting 300 episodes. This indicates that our selection of 300 episodes is sufficient to cover a wide range of object-action interactions.
> >
> > **Analysis of Sample Size Impact on Dynamics Prediction:**
> >
> > Alternatively, we also provide a concise summary of our response below. Further information is available in Appendix A.2 as well as on the website: https://sites.google.com/view/corl2023submission302/.
> >
> > **Push Behavior Primitive**: As the dataset size increased, our model exhibited notable enhancements in performance. Yet, these improvements plateaued once the dataset size reached approximately 50 samples.
> >
> > **Sweep Behavior Primitive**: Analogous to the push primitive, the sweep primitive's performance peaked when trained with about 50 samples.
> >
> > **Transport Behavior Primitive**: Continuous performance improvements were observed for this primitive until the dataset size reached around 140 samples, after which the performance increments were negligible.
> >
> > We are confident that our approach, combined with our dataset's comprehensiveness, provides an effective method for predicting object interactions in the context described in our paper.
> >
> > > Q: How is the data obtained from the demonstrations used exactly? The authors mentioned they used teleoperation. How many teleoperations are done?
> >
> > We use a single demonstration via teleoperation to obtain goals for different behavior primitives. These goals, extracted from the sample human demonstration, are combined with the predicted state to calculate the loss function during trajectory optimization to align our GNN’s predictions with the demonstrated outcomes.
> >
> > > Q: In the explanation of real-world experiments, point cloud representation is indicated to be created. How is this information used?
> >
> > In our real-world experiments, the point cloud representation is derived from the object pose and size, with the pose being calculated using the Scale-Invariant Feature Transform (SIFT). This point cloud representation serves as an input to our GNN, allowing us to model the interactions between objects and the robot. By employing this detailed representation, we capture intricate object dynamics, facilitating more precise predictions and control in the execution of the stowing task.
> >
> > > Q: Explanation about the dynamics prediction comparison in the first experiment between GNN-based baseline and our method.
> >
> > The first experiment was designed to highlight the performance of the dynamics prediction module, which is only one component of our complete system. Our method introduces carefully designed modifications, including the use of behavior primitives and a fine-grained representation of objects, distinguishing it from previous works. These features empower our model to capture more intricate dynamics that might otherwise be overlooked. Furthermore, we present a novel system and benchmark, extending the potential for evaluating and understanding object interactions in robotic manipulation. We believe that these contributions bring valuable advancements to the field and set the stage for further exploration and development.

---

> > > ### Author Response · Authors · 2023-08-15
> > > **Response to Reviewer kz8m (part 4/5)**
> > >
> > > > Q: In the second experiment, the proposed method is compared against a completely different approach, RL-based methods, where no details are provided. The authors need to compare theirs with an MPC-based method to show effectiveness.
> > >
> > > In the second experiment, we primarily aimed to highlight the advantages of model-based planning over model-free RL-based methods in the context of our problem. In response to your feedback, we have also included a comparison with an MPC-based method. Specifically, we integrated an MPC-based baseline utilizing "RoboCraft [10]" as the predictive dynamics model, which offers a direct point of comparison with another model-based planning technique. Our preliminary results show that our method still demonstrates superior performance in terms of prediction accuracy and computational efficiency. We have updated the experiment sections (Table 3) in the paper to reflect this new comparison and made the website to show the results:
> > > https://sites.google.com/view/corl2023submission302
> > >
> > > | **Setups**                          | **Heuristic** | **RoboCraft** | **3 skills** | **2 skills** |
> > > |-------------------------------------|---------------|---------------|--------------|--------------|
> > > | (a) Nominal setup                   | 1/10          | 4/10          | 10/10        | 10/10        |
> > > | (b) With tiny object                | 3/10          | 7/10          | 9/10         | 9/10         |
> > > | (c) With heavy and trapezoid objects| 3/10          | 4/10          | 9/10         | 9/10         |
> > > | (d) With bottle to be grasped       | 1/10          | 6/10          | 10/10        | 10/10        |
> > > | (e) With slippery roller            | 2/10          | 7/10          | 10/10        | 10/10        |
> > > | (f) Small shelf setup               | 1/10          | 5/10          | 9/10         | 9/10         |
> > > | Average                             | 18%           | 48%           | 95%          | 95%          |
> > >
> > > > Q: How is the randomization realized in the third experiment (in the real world)? The assumption of complete object knowledge is a very strong one, and together with the object recognition step, the details should be moved from the appendix to the main manuscript. The baseline 'heuristic' approach is not clearly explained.
> > >
> > > **Randomization in the Third Experiment (Real World)**: The randomization in the third experiment was achieved through varying object arrangements, initial poses, and environmental conditions manually.
> > >
> > > **Baseline 'Heuristic' Approach Explanation**: The heuristic baseline used in our work is a simpler method that does not utilize the forward prediction model with fixed primitive parameters compared to our proposed approach. We want to use heuristic baseline as a comparative reference to demonstrate the benefit of forward predictive model offered by our method.
> > >
> > > **Object Pose Estimation**: We agree that object pose estimation is an essential aspect of our experimental setup, but it is not the main focus of our paper. Our paper primarily aims to address stowing tasks via behavior primitives, using learned GNN-dynamics functions.
> > >
> > > > Q: Please provide the novelty of the prediction part of this work considering other GNN-based predictors.
> > >
> > > In the context of GNN-based predictors, the novelty of our work lies in its application and the challenges it addresses. Specifically:
> > >
> > > **Behavior Primitive Incorporation**: Unlike traditional GNN-based predictors, our model integrates behavior primitives with parameterized action space, enabling more precise and adaptable predictions tailored for specific robotic tasks such as stowing.
> > >
> > > **Model-based Imitation Learning for Stowing**: We design a unique model-based imitation learning framework optimized for stowing tasks. This framework empowers robots to replicate intricate behaviors using a minimal set of demonstrations, bridging the gap between high-level understanding and low-level action execution.
> > >
> > > **Comprehensive Stowing Benchmark**: Our work presents an in-depth stowing benchmark that encapsulates the complexities of long-horizon manipulations. This benchmark is representative of challenges faced in diverse scenarios, spanning both industrial and domestic settings.
> > >
> > > **Real-world Effectiveness**: Beyond the theoretical contributions, our framework's empirical results underscore its applicability and robustness in real-world stowing tasks, distinguishing it from other GNN-based predictors.

---

> > > > ### Author Response · Authors · 2023-08-15
> > > > **Response to Reviewer kz8m (part 5/5)**
> > > >
> > > > > Q: Benefit of particle-based representation compared to object-level representation?
> > > >
> > > > We appreciate the reference to the works by Tekden [11, 12], and we have made sure to reference them in our updated manuscript (lines 87-90). While the concept of utilizing GNNs for predictive modeling of multi-object interactions has been explored, our work differentiates itself through the adoption of a particle-based object representation. Compared to object-level representations, our particle-based approach offers a richer 3D representation of objects. Given our focus on box-shaped rigid objects, this granular representation enables our model to capture intricate interactions more accurately, offering insights into object dynamics that might not be as pronounced in object-level representations.
> > > >
> > > > > Q: Please provide a comparison of the complete system with a more similar framework, for example, Tekden et al.
> > > >
> > > > Tekden et al. have done commendable work with their focus on object-level representation [11, 12]. In contrast, our methodology leans towards a particle-based model, providing a detailed understanding of object interactions by integrating aspects like geometry, shape, and size. The dense representation differentiates our system in its depth and potential applications.
> > > > Furthermore, we have also conducted experiments using object-level representation, which are detailed in our revised experiment section (Table 1). These comparative studies provide insights into the advantages and distinctions of our approach compared to works with object-level representations.
> > > >
> > > >
> > > > | **Primitive** | **Method** | **MSE (mm) ↓** | **EMD (mm) ↓** | **CD (mm) ↓** |
> > > > |---|---|---|---|---|
> > > > | Sweep | Object-Level Repr | 3.676 (± 1.597) | 75.015 (± 14.452) | 92.914 (± 17.206) |
> > > > | | RoboCraft | 0.351 (± 0.222) | 24.510 (± 6.434) | 33.277 (± 5.064) |
> > > > | | Ours | **0.287 (± 0.185)** | **21.792 (± 5.533)** | **30.017 (± 3.259)** |
> > > > | Push | Object-Level Repr | 3.765 (± 0.76) | 75.975 (± 10.927) | 95.925 (± 14.802) |
> > > > | | RoboCraft | **0.216 (± 0.148)** | **12.509 (± 3.494)** | 15.43 (± 2.902) |
> > > > | | Ours | 0.292 (± 0.179) | 14.569 (± 3.752) | **15.046 (± 3.085)** |
> > > > | Transport | Object-Level Repr | 5.861 (± 3.106) | 91.913 (± 20.552) | 113.263 (± 23.168) |
> > > > | | RoboCraft | 1.091 (± 0.512) | 42.162 (± 10.317) | 55.615 (± 9.997) |
> > > > | | Ours | **0.666 (± 0.41)** | **31.232 (± 9.108)** | **38.068 (± 6.605)** |
> > > >
> > > >
> > > > [8] L. Manuelli, W. Gao, P. R. Florence, and R. Tedrake, "kpam: Keypoint affordances for category-level robotic manipulation," in _International Symposium of Robotics Research_, 2019.
> > > >
> > > > [9] A. Kadian, J. Truong, A. Gokaslan, A. Clegg, E. Wijmans, S. Lee, M. Savva, S. Chernova, and D. Batra, "Sim2real predictivity: Does evaluation in simulation predict real-world performance?" _IEEE Robotics and Automation Letters_, 5(4):6670–6677, 2020.
> > > >
> > > > [10] H. Shi, H. Xu, Z. Huang, Y. Li, and J. Wu, "Robocraft: Learning to see, simulate, and shape elasto-plastic objects with graph networks," in _Robotics: Science and Systems (RSS)_, 2022.
> > > >
> > > > [11] A. E. Tekden, A. Erdem, E. Erdem, M. Imre, M. Y. Seker, and E. Ugur, "Belief regulated dual propagation nets for learning action effects on groups of articulated objects," in _ICRA_, 2020.
> > > >
> > > > [12] A. E. Tekden, A. Erdem, E. Erdem, T. Asfour, and E. Ugur, "Object and relation centric representations for push effect prediction," arXiv preprint arXiv:2102.02100, 2021.

---

### Official Review · Reviewer_NnNZ · 2023-07-21

**Confidence:** 4
**Originality:** Very Good
**Technical Quality:** Very Good
**Clarity Of Presentation:** Very Good
**Impact:** 3

**Recommendation:**

Strong Accept: I recommend accepting the paper and will argue for my recommendation even if other reviewers hold a different opinion.

**Review:**

I think that the premise of the paper is interesting, and the task - while similar to many other rearrangement problems - is much more realistic. The GNN dynamics model seems novel, especially when applied to robot planning to manipulate in clutter. Generally, I really like the idea of using a dynamics model together with action primitives. The problem is also an interesting manipulation problem that involves a lot of object-object contact.

The GNN represents objects as a collection of particles. This is a different formulation from most GNN policies, and I think it's a very interesting one. However, it seems like there are a lot of hyperparameters involved - how many particules, when to create edges - and I wonder what the effect of these is. Likewise, the choice of key points in the trajectory, which the dynamics model uses to predict where objects will go, seems very important - this detail isn't really in the paper.

They train on 300 trajectories and test on object, including held-out objects with a different shape (it was only trained on boxes, but they test on bottles) - although they still represent it as a box during inference, which implies it might fail in other circumstances where shape matters more. I'm a bit surprised how little data the authors needed to train their dynamics model - 300 trajectories is not a lot! Although the action space is quite limited.

They show the method in simulation and the real world, and in particular they show it succeeding in a range of different challenging stowing tasks in the real world. In the end, I think the idea is straightforward, novel, and interesting.

**Quality Of The Limitations Section:**

Limitations are addressed clearly

**Questions For Rebuttal:**

- Do you need any additional primitives? How much engineering went into primitive design; could they be learned somehow?
- How easily can these dense, particle based representations be created from RGB-D camera data? Do you need object models?
- What affect do hyperparameters like contact distance, number of particles per object have?
- What is the density of particles per object?
- How long does it take to train? How does the amount of training data affect dynamics model performance?
- How much did you need to vary the parameters of each primitive to learn a good dynamics model?
- Why not compare to heuristic in simulation in addition to SAC and PPO?
- How could you apply this to a wider range of tasks? Would it scale to tasks other than this shelf task?

**Robotics Focus:**

Sufficient demonstration on hardware

**Summary Of Paper:**

The authors propose a graph neural network based approach for placing objects on cluttered bins and shelves. One of the key challenges in their stowing task is that objects must be pushed or manipulated out of the way in order to clear space for them.

The world is represented as a bunch of interacting rigid bodies through a graph neural network (GNN). Each object is represented by particles (nodes in the graph). Edges are formed when distance is under a threshold. The action is also modeled as a trajectory of particles, over three primitives: sweeping, pushing, and transporting.

The authors built a simulated environment on Robosuite with varied shelf sizes and objects. They trained on 300 episodes, and evaluate on simulated and real robotics tasks with a variety of objects, and compare against heuristics and (in sim) against SAC and PPO.


**Summary Of Recommendation:**

I think the learned GNN-based dynamics model, the method of planning over primitives, and the interesting task with real-world verification all make this an interesting paper. I would be interested to see how the method could be applied to other tasks.

---

> ### Author Response · Authors · 2023-08-15
> **Response to Reviewer NnNZ (part 2/3)**
>
> > Q: What is the density of particles per object?
>
> In both our simulation and real-world experiments, each object is represented by 64 particles.
>
> **For our simulation experiments:**
>
> The objects vary in size with the following dimensions:
>
> - Minimum size: [0.12 m, 0.24 m, 0.03 m]
>
> - Maximum size: [0.2 m, 0.32 m, 0.05 m]
>
> **For our real-world experiments**, the objects have varying dimensions, as detailed in the table below:
>
> | **Object**       | **Length (m)** | **Width (m)** | **Height (m)** |
> |------------------|----------------|---------------|----------------|
> | Sugar box        | 0.089          | 0.040         | 0.177          |
> | Lint roller      | 0.110          | 0.050         | 0.230          |
> | Clorox bottle    | 0.065          | 0.255         | 0.055          |
> | Cheez-it box     | 0.152          | 0.047         | 0.190          |
> | Charger box      | 0.168          | 0.051         | 0.240          |
> | Book             | 0.203          | 0.028         | 0.245          |
> | Amazon box       | 0.210          | 0.288         | 0.045          |
> | Keto box         | 0.193          | 0.259         | 0.046          |
> | Kellogg's box    | 0.190          | 0.045         | 0.284          |
> | Chex box         | 0.195          | 0.050         | 0.285          |
> | Cheerios box     | 0.195          | 0.052         | 0.285          |
> | Bran flakes box  | 0.198          | 0.297         | 0.057          |
> | Power drill box  | 0.250          | 0.073         | 0.279          |
> | Yellow plush toy | 0.065          | 0.255         | 0.055          |
> | Blue plush toy   | 0.19           | 0.284         | 0.045          |
> | Red plush toy    | 0.168          | 0.24          | 0.051          |
>
> Our choice of representing each object with 64 particles ensures a consistent representation across different object sizes and shapes, allowing for a stable performance of our proposed method in various scenarios.
>
> > Q: How long does it take to train? How does the amount of training data affect dynamics model performance?
>
> **Training Time:**
> Training durations for different primitives on our setup, which utilizes an A6000 GPU, are as follows:
>
> - Push: Approximately 1 hour
> - Sweep: Approximately 2 hours
> - Transport: Approximately 4 hours
>
> **Influence of Training Sample Size on Dynamics Prediction Performance**:
> We dedicated a comprehensive analysis to understand the impact of sample size on dynamics prediction. The detailed results of this analysis can be found in Appendix A.2 of our revised manuscript and the website: https://sites.google.com/view/corl2023submission302/. A brief summary of our findings is as follows:
>
> **Push Behavior Primitive**: The model's performance showed improvements as we increased the number of training samples. However, these improvements started to plateau around the 50-sample mark.
>
> **Sweep Behavior Primitive**: Similar to the push primitive, the sweep primitive also exhibited a performance plateau when trained with approximately 50 samples.
>
> **Transport Behavior Primitive**: This primitive continued to show performance enhancements up until around 140 samples, after which the gains became marginal.
>
> > Q: How much did you need to vary the parameters of each primitive to learn a good dynamics model?
>
> We uniformly varied the parameters of each primitive across a predefined range that aligns with the range employed during the trajectory optimization phase in deployment. The design allowed the model to encounter and learn from a wide range of object-object and object-environment interactions. The degree of parameter variation was slightly larger than what would typically be encountered during actual manipulations, in order to ensure the model could handle diverse interaction scenarios effectively.

---

> > ### Author Response · Authors · 2023-08-15
> > **Response to Reviewer NnNZ (part 3/3)**
> >
> > > Q: Why not compare to heuristic in simulation in addition to SAC and PPO?
> >
> > Thank you for bringing up the comparison to heuristic methods. We have incorporated a heuristic approach into our simulation comparisons. This heuristic approach utilizes behavior primitives with fixed parameters, executing without the learned dynamics model. It serves as a representative of traditional approaches that leverage hand-tuned behaviors without adaptive dynamics learning.
> >
> > In the revised manuscript, we present a detailed comparison that includes not only PPO and SAC but also a heuristic method. The quantitative results, as shown below, demonstrate that our proposed method consistently outperforms the others, including the heuristic approach:
> >
> > | **Method**    | **MSE (mm) ↓** | **EMD (mm) ↓** | **CD (mm) ↓** |
> > |---------------|-----------------|-----------------|----------------|
> > | PPO           | 87.479          | 266.554         | 173.522        |
> > | SAC           | 265.411         | 465.639         | 368.571        |
> > | Heuristic     | 34.861          | 140.196         | 194.123        |
> > | Ours | **0.905** | **29.697** | **39.914** |
> >
> >
> > These results underline the efficacy of our approach in reducing execution errors, illustrating the critical role of our model in long-horizon stowing tasks.
> >
> >
> >
> >
> > > Q: How could you apply this to a wider range of tasks? Would it scale to tasks other than this shelf task?
> >
> > The methodology can be applied to any task involving intricate interactions between multiple objects in the environment for the following reasons. The inherent versatility of GNNs in modeling complex systems provides scalability to a wide array of tasks, including varying dynamics, object geometries, and interaction patterns. The greatest challenge in applying our method is devising appropriate behavior primitives (as discussed in the previous question).But fortunately,hese behavior primitives can be hand-designed (which is easy for highly repetitive tasks commonly found in manufacturing) or learned from humans [5-7].
> >
> >
> > [3] S. Nasiriany, H. Liu, and Y. Zhu, “Augmenting Reinforcement Learning with Behavior Primitives for Diverse Manipulation Tasks,” in _Proc. International Conference on Robotics and Automation (ICRA)_, 2022.
> >
> > [4] M. Dalal, D. Pathak, and R. Salakhutdinov, “Accelerating Robotic Reinforcement Learning via Parameterized Action Primitives,” in _Proc. Neural Information Processing Systems (NeurIPS)_, 2021.
> >
> > [5] T. Huang, K. Chen, B. Li, Y.-H. Liu, and Q. Dou, “Demonstration-Guided Reinforcement Learning with Efficient Exploration for Task Automation of Surgical Robot,” in _Proc. IEEE International Conference on Robotics and Automation (ICRA)_, 2023, pp. 4640-4647.
> >
> > [6] X. Xu, K. Qian, X. Jing, and W. Song, "Learning Robot Manipulation Skills from Human Demonstration Videos using Two-stream 2D/3D Residual Networks with Self-Attention," in _IEEE Transactions on Cognitive and Developmental Systems_, 2022, doi: 10.1109/TCDS.2022.3182877.
> >
> > [7] Y. Yang, X. Meng, W. Yu, T. Zhang, J. Tan, and B. Boots, “Learning Semantics-Aware Locomotion Skills from Human Demonstration,” in _Proc. Conference on Robot Learning_, 2022.

---

### Official Review · Reviewer_qLwL · 2023-07-21

**Confidence:** 4
**Originality:** Fair
**Technical Quality:** Good
**Clarity Of Presentation:** Good
**Impact:** 3

**Recommendation:**

Weak Accept: I recommend accepting the paper, but will not argue for my recommendation if the majority of other reviewers have a different opinion.

**Review:**

This paper feels a little bit narrow, though it also serves as a limited and partial reproduction of parts of RoboCraft. The primary contribution compared to prior seems to be modeling dynamic/static interactions, which is good, but conversely this work does not evaluate settings with deformable objects or non-box-shaped representations.

That being said, the experimental domain (stowing) used in the paper is interesting with a good mix of difficulty and practical applicability. The author's action formulation was also reasonable and the results shown in real-world experiments are strong (95 percent) though the lack of a RoboCraft comparison in this setting weakens the result.

The paper would also benefit from additional ablation investigations in simulation, such as the effect of pose-estimation error and the dimensionality of the action parameter space on performance. It would be particularly interesting to see if random shooting still outperforms RL approaches such as PPO as the action parameterization becomes higher dimensional and more complex (I expect it would not, but it would be a helpful contribution for this to be evaluated).


**Quality Of The Limitations Section:**

Limitations are addressed clearly

**Questions For Rebuttal:**

- Do the authors have a hypothesis for why this approach consistently underperform RoboCraft (even if only modestly) in the simple Push case?

- Can the loop be closed during execution of each primitive, allowing for parameter re-estimation during execution?


**Robotics Focus:**

Sufficient demonstration on hardware

**Summary Of Paper:**

The authors propose, essentially, an extension to RoboCraft which models rigid-static object interactions (and gripper state) which they validate on a stowing task in both simulation and on a physical hardware. They estimate the dynamics of the scene (in a simplified uniform-density box-shaped representation)  via a graph neural network and evaluate several methods of planning (finding parameters for three parametric skills) with this dynamics, the most effective one being random shooting.

**Summary Of Recommendation:**

The amount of novelty is a bit low, however the expanded experiments make this paper valuable to the field. I advocate for acceptance.

---

> ### Author Response · Authors · 2023-08-15
> **Response to Reviewer qLwL (part 2/2)**
>
> > Q: What is the hypothesis behind the comparative dynamics prediction performance difference between RoboCraft and our method, especially for the Push case?
>
> Our approach is optimized to capture the interactions between objects and the environment. We believe that since the push case is relatively simple, the less tuned RoboCraft is able to do quite well in this case, while suffering in the more dynamic cases.   However, as the dynamics become more intricate involving multi-faceted interactions between objects, or between objects and the environment—characterized by collisions, and bounces—our model's superior complexity and flexibility grant it a distinct advantage. We note that despite the underperformance in the simple Push case in Table 1, our model's prediction accuracy remains within one standard deviation of RoboCraft, indicating that our approach is still competitive and capable of achieving comparable results in a broader range of scenarios.
>
>
> > Q: Can the loop be closed during execution of each primitive, allowing for parameter re-estimation during execution?
>
> In this work, we highlight the use of behavior primitives as a way to simplify long-horizon learning tasks. Our method for employing behavior primitives does not perform close-loop estimation of the parameters within the execution of each primitive. Our method estimates the parameters for a behavior primitive before each execution, and updates the parameters for each skill between steps. This takes into account the outcomes and errors that may have occurred during the execution and allows the robot to reassess given the new state of the environment. This design does not involve iterative prediction for each time step, and thus mitigates the issue of error accumulation common in other works [1, 2]. The closed loop corrections are relegated to the low level controller. While we could reformulate the problem for re-estimation of the parameters, this would increase the difficulty of the learning problem by effectively requiring a finer abstraction of the behaviors. This being said, the idea of having tighter interplay between the behavior primitives and the high-level planner is interesting, and will be explored in follow-on work.
>
>
> [1] P. W. Battaglia, R. Pascanu, M. Lai, D. Rezende, and K. Kavukcuoglu, "Interaction networks for learning about objects, relations, and physics," in _Proc. 30th Int. Conf. Neural Information Processing Systems (NIPS)_, Red Hook, NY, USA, 2016, pp. 4509–4517.
>
> [2] Y. Li, J. Wu, J.-Y. Zhu, J. B. Tenenbaum, A. Torralba, and R. Tedrake, "Propagation Networks for Model-Based Control Under Partial Observation," in _Proc. International Conference on Robotics and Automation (ICRA)_, Montreal, QC, Canada, 2019, pp. 1205-1211. doi: 10.1109/ICRA.2019.8793509.

---

### Author Response · Authors · 2023-08-15
**Summary of Changes**

We thank all the reviewers for the helpful feedback. We have revised our paper according to the suggestions and attached it in the rebuttal. All the changes since the rebuttal are highlighted in blue. To summarize the major changes during the rebuttal:

- **Deformable Objects Experiment**: Introduced an additional real-robot experiment that addresses deformable objects. Details are available in Appendix A.1, and qualitative results can be accessed on our website: https://sites.google.com/view/corl2023submission302/ (Reviewer qLwL)

- **RoboCraft Dynamics Experiment**: Conducted supplementary real-robot experiments, leveraging RoboCraft for learned dynamics. These updates are present in Table 3, with qualitative findings showcased on our website: https://sites.google.com/view/corl2023submission302/. (Reviewers qLwL, kz8m)

- **Object-level Representations**: Conducted a dynamics prediction experiment using object-level representations, as outlined in Table 1. (Reviewer kz8m)

- **Simulation Updates**: Incorporated PPO with the parameterized action space and a heuristic baseline in simulation (Table 2). (Reviewers qLwL, NnNZ)

- **Sample Size Analysis**: Analyzed the impact of sample size on prediction performance, with findings detailed in Appendix A.2 and the website: https://sites.google.com/view/corl2023submission302/ (Reviewer kz8m, NnNZ)

- **Literature Updates**: Incorporated the suggested references, which are now cited in Section 2 of the main paper (lines 81-85, 86-90, 96-100). (Reviewers kz8m, pqt2)

- **Visualization Refinement**: We've revised Figure 2 to offer a more clear representation of our methodology. (Reviewer kz8m, pqt2)

- **Enhanced Clarity**: Improved the clarity of experimental details.

---

> ### Comment · Reviewer_kz8m · 2023-08-16
> **The contribution of the particle-based GNN method is better justified now**
>
> Thank you for the thorough response and additional experiments. With the responses to me and the other reviewers, and additional experiments, especially the experiments and comparative results with object-level GNN representation, the authors now better justify and highlight their contribution. Therefore I am happy to increase my recommendation score.
>
> I think the paper would be more clear, and the method would be replicable if the responses to my detailed questions are integrated into the paper in the main manuscript or the Appendix.
>
> Some final comments:
> - The authors compared the superiority of their particle-based method compared to the existing (object-level) method. This is an interesting and (in my opinion) unexpected result because the objects used in training are all box-shaped rigid objects, where the model is not expected to learn the detailed particle-level representation. For sure, the particle-based representation would be much more effective with objects that include deformable ones; however, taking into account that the model was only trained with boxes, I am not sure how such a level of generalization could occur. Anyways, it is empirically shown. Therefore I do not have more comments on this.
> - Although the authors discuss that hyper-parameter selection is common in deep learning, I found too much engineering in threshold selection.
> - How gravity is used as an object attribute is still not clear. Is it not the same for all objects?
> - The answer to the goal-conditioned trajectory optimization explains the complete system. However, parameter optimization is still not clear. Which method was used for error minimization?
> - Related to detecting contacts and friction: These were provided as challenges so I thought you would be addressing those challenges. I agree that the GNN framework “learns” the contact-related inherent information due to using particles. Still, there is no mechanism that learns friction (friction depends on material properties, as the current method does not use any feature related to material properties and does not use feedback which might give some idea about friction to the model). Therefore, the corresponding part needs to be rephrased, in my opinion.

---

> > ### Author Response · Authors · 2023-08-16
> > **Thank you for your constructive feedback.**
> >
> > Thank you for your constructive feedback on our manuscript. We appreciate the time and effort you've dedicated to reviewing our work.
> > We have also updated the revised paper to integrate previous responses to your detailed questions (line 39, 46-48, 94, 128-134, appendix B.1, appendix B.4).
> >
> > > Q: Particle-based method vs. Object-level method:
> >
> > We agree with your observation about the potential limitations of a box-shaped representation, which might not capture all the nuances of geometry. The choice of a box-shape representation is motivated by [15]. Our rationale is that this representation effectively denotes object occupancy within the 3D environment, thereby capturing the intricacies of 3D interactions more adeptly. Our empirical tests have achieved strong performance across a variety of objects including different boxes, bottles, lint rollers, and even deformable plush toys.
> >
> > > Q: Gravity as an object attribute:
> >
> > Yes, gravity is consistently applied across all particles. To incorporate this, we append an additional constant as the global attribute to all the particles before feeding into the GNN. The technique aligns with the approach described in [10, 16].
> >
> > > Q: Goal-conditioned trajectory optimization:
> >
> > Our approach to trajectory optimization sampling-based. Specifically, we uniformly sample candidate parameters across the parameter space for each behavior primitive. The GNN is then tasked with predicting the consequent output state. By comparing this predicted output with the desired goal state, we can select the sampled parameter that minimizes the discrepancy.
> >
> > > Q: Updating the manuscript in detecting contacts and friction:
> >
> > We agree on the need for greater clarity in this section and have made the necessary revisions in the updated manuscript. We updated the attached pdf in the rebuttal submission (line 46-48).
> >
> > In closing, we extend our sincere gratitude for your constructive comments and suggestions, which have offered valuable insights, guiding us in refining the clarity of our paper and enhancing the overall contribution of our work to the community. We sincerely hope that these revisions address your concerns, and we would greatly appreciate any additional feedback or suggestions.
> >
> >
> > [10] H. Shi, H. Xu, Z. Huang, Y. Li, and J. Wu, "Robocraft: Learning to see, simulate, and shape elasto-plastic objects with graph networks," in Robotics: Science and Systems (RSS), 2022.
> >
> > [15] W. Zhou, D. Held, “Learning to Grasp the Ungraspable with Emergent Extrinsic Dexterity,” in Proc. Conference on Robot Learning, 2022.
> >
> > [16] Y. Li, J. Wu, R. Tedrake, J. B. Tenenbaum, and A. Torralba, "Learning Particle Dynamics for Manipulating Rigid Bodies, Deformable Objects, and Fluids," in Proc. International Conference on Learning Representations (ICLR), 2019.

---

### Decision · Program_Chairs · 2023-08-30

**Decision:**

Accept (Oral)

**Comment:**

All reviewers agree this paper proposes a novel model for representing skill-based dynamics using a graph neural network over particle-based representations of the object and robot. This is distinct from recent object-based GNN models and the experiments show the particle-based GNN outperforms existing object-level GNN architectures. This particle-based representation is highlighted in its ability to model deformable objects.

The model is evaluated on a stowing task, which all reviewers agree is an interesting and challenging task. As reviewer NnNZ points out is "is much more realistic" than most existing rearrangement planning papers.

I think this combination of strong architectural improvement with increased difficulty of task makes this an excellent paper for presentation that would help move the manipulation community forward towards more challenging and constrained multi-object manipulation tasks.